# ROPUST: Improving Robustness through Fine-tuning with Photonic Processors and Synthetic Gradients

Alessandro Cappelli [1]   Julien Launay [1 2]   Laurent Meunier [3 4]   Ruben Ohana [1 2]   Iacopo Poli [1]

## Abstract

Robustness to adversarial attacks is typically obtained through expensive adversarial training with Projected Gradient Descent. We introduce RO-PUST, a remarkably simple and efficient method to leverage robust pre-trained models and further increase their robustness, at no cost in natural accuracy. Our technique relies on the use of an Optical Processing Unit (OPU), a photonic co-processor, and a fine-tuning step performed with Direct Feedback Alignment, a synthetic gradient training scheme. We test our method on nine different models against four attacks in Robust-Bench, consistently improving over state-of-the-art performance. We also introduce phase retrieval attacks, specifically designed to target our own defense. We show that even with state-of-the-art phase retrieval techniques, ROPUST is effective.

## 1. Introduction

Adversarial examples (Goodfellow et al., 2015) threaten the safety and reliability of machine learning models deployed in the wild. Because of the sheer number of attack and defense scenarios, robustness can be difficult to evaluate (Bubeck et al., 2019). Standardized benchmarks, such as RobustBench (Croce et al., 2020) with AutoAttack (Croce & Hein, 2020b), have helped better evaluate progress in the field. The development of defense-specific attacks is also crucial (Tramèr & Boneh, 2019). To date, one of the most effective defense techniques remains adversarial training with Projected Gradient Descent (PGD) (Madry et al., 2018). Adversarial training can be resource-consuming, but robust networks pre-trained with PGD are now widely available, motivating their use as a foundation for simple and widely

[1]LightOn, France [2]LPENS, École Normale Supérieure, France [3]Facebook AI Research, France [4]Université Paris-Dauphine, France. Correspondence to: Alessandro Cappelli <alessandro@lighton.ai>.

*Accepted by the ICML 2021 workshop on A Blessing in Disguise: The Prospects and Perils of Adversarial Machine Learning.* Copyright 2021 by the author(s).

applicable defenses that further enhance their robustness.

To this end, we introduce **ROPUST**, a drop-in replacement for the classifier of already robust models. Our defense leverages a photonic co-processor (the Optical Processing Unit, OPU) for physical *parameter obfuscation* (Cappelli et al., 2021): because the *fixed* random parameters are optically implemented, they remain unknown at training and inference time. Additionally, a synthetic gradient method, Direct Feedback Alignment (DFA) (Nøkland, 2016), is used to fine-tune the ROPUST classifier.

We evaluate our method against AutoAttack on nine different models in RobustBench, and consistently improve robust accuracies over the state-of-the-art (Fig. 1). We also develop a *phase retrieval* attack targeting our parameter obfuscation, and show that ROPUST remains effective.

### 1.1. Related work

**Attacks.** Adversarial attacks have been framed in a variety of settings: white-box, where the attacker is assumed to have unlimited access to the model, including its parameters (e.g. FGSM (Goodfellow et al., 2015), PGD (Madry et al., 2018; Kurakin et al., 2016), Carlini & Wagner (Carlini & Wagner, 2017)); black-box, assuming only limited access to the network for the attacker, with methods attempting to estimate the gradients (Chen et al., 2017; Ilyas et al., 2018a;b), or derived from genetic algorithms (Andriushchenko et al., 2019; Meunier et al., 2019) and combinatorial optimization (Moon et al., 2019); transfer attacks, where an attack is crafted on a model that is accessible to the attacker, and then applied to the target network (Papernot et al., 2016). Automated schemes, such as AutoAttack (Croce & Hein, 2020b), have been proposed to autonomously select attacks and tune their hyperparameters.

**Defenses.** Adversarial training adds adversarial robustness as an explicit training objective (Goodfellow et al., 2015; Madry et al., 2018), by incorporating adversarial examples during the training. Theoretically grounded defenses have been proposed (Lecuyer et al., 2018; Cohen et al.; Alexandre Araujo & Negrevergne, 2020; Pinot et al., 2019; Wong et al., 2018; Wong & Kolter, 2018), but these fail to match the clean accuracy of state-of-the-art networks. Many empir-

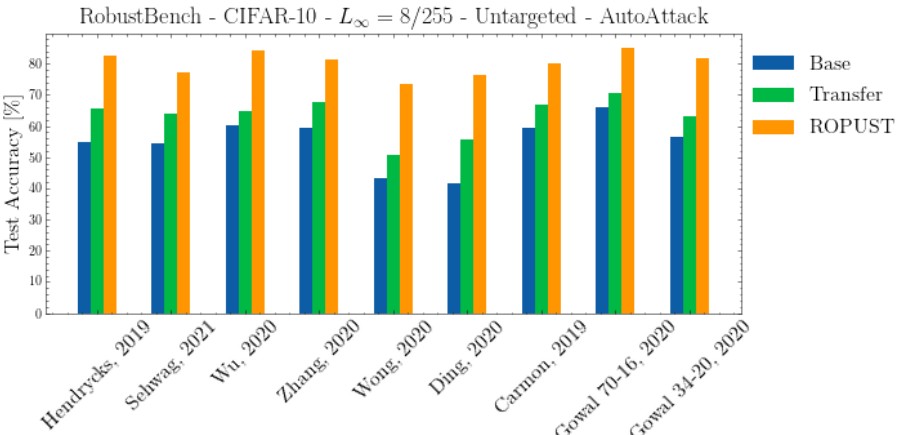

*Figure 1.* **ROPUST systematically improves the test accuracy of already robust models**. Transfer refers to the performance when attacks are generated on the base model and transferred to the ROPUST model. Models from the RobustBench model zoo: Hendrycks, 2019 (Hendrycks et al., 2019), Sehwag, 2021 (Sehwag et al., 2021), Wu, 2020 (Wu et al., 2020), Zhang, 2020 (Zhang et al., 2020), Wong, 2020 (Wong et al., 2020), Ding, 2020 (Ding et al., 2020), Carmon, 2019 (Carmon et al., 2019), Gowal, 2020 (Gowal et al., 2020).

ical defenses have been criticized for providing a false sense of security (Athalye et al., 2018; Tramèr & Boneh, 2019), by not evaluating on attacks adapted to the defense. Gradient obfuscation, through the use of a non-differentiable activation function, has been proposed as a way to protect against white-box attacks (Papernot et al., 2017). However, it can be easily bypassed by Backward Pass Differentiable Approximation (BPDA) (Athalye et al., 2018), where the defense is replaced by a differentiable relaxation. *Parameter obfuscation* has been proposed with dedicated photonic co-processor (Cappelli et al., 2021). However, by itself, this kind of defense falls short of adversarial training.

**Fine-tuning and analog computing.** Previous work introduced *adversarial fine-tuning* (Jeddi et al., 2020): fine-tuning a non-robust model with an adversarial objective. In this work instead we fine-tune a robust model without adversarial training. Additionally, it was shown that robustness improves transfer performance (Salman et al., 2020) and that robustness transfers across datasets (Shafahi et al., 2020). The advantage of non-ideal analog computations in terms of robustness has been investigated in the context of NVM crossbars (Roy et al., 2020).

### 1.2. Motivations and contributions

We propose to simplify and extend the applicability of photonic-based parameter obfuscation defenses. The use of dedicated hardware to perform the random projection physically guarantees *parameter obfuscation* (Cappelli et al., 2021). Our defense, ROPUST, can be dropped-in to supplement any robust pre-trained model and fine-tuning its classifier is fast. In contrast with existing parameter-

obfuscation methods, it leverages pre-trained robust models, and achieves state-of-the-art performance. Drawing inspiration from the field of phase retrieval, we introduce a new kind of attack against defenses relying on parameter obfuscation, *phase retrieval attacks*. We show that ROPUST remains robust even against state-of-the-art retrieval methods.

## 2. Methods

### 2.1. Automated adversarial attacks

We evaluate our model against the four attacks implemented in RobustBench: APGD-CE and APGD-T (Croce & Hein, 2020b), Square attack (Andriushchenko et al., 2019), and Fast Adaptive Boundary (FAB) attack (Croce & Hein, 2020a). We describe these attacks more in detail in the supplementary. In RobustBench, using AutoAttack, given a batch of samples, these are first attacked with APGD-CE. Then, the samples that were successfully attacked are discarded, and the remaining ones are attacked with APGD-T. This procedure continues with Square and FAB attack.

### 2.2. Our defense

**Optical Processing Units.** Optical Processing Units (OPU)[1] are photonic co-processors dedicated to efficient large-scale random projections (Ohana et al., 2020). Assuming an input vector $\mathbf{x}$, the OPU computes the following operation using light scattering through a diffusive medium:

$$\mathbf{y} = |\mathbf{U}\mathbf{x}|^2 \tag{1}$$

---

[1]Accessible at https://cloud.lighton.ai.

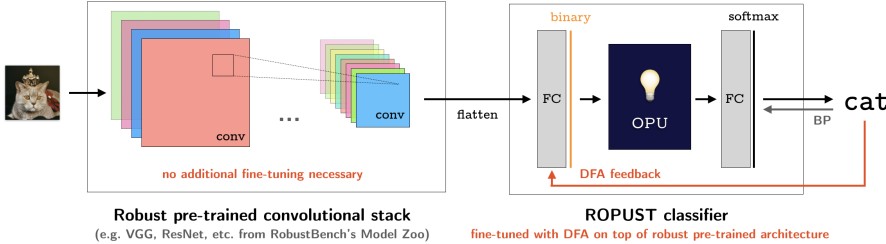

Figure 2. **ROPUST replaces the classifier of already robust models, enhancing their adversarial robustness.** Only the ROPUST classifier needs fine-tuning; the convolutional stack is frozen. Convolutional features first go through a fully-connected layer, before binarization for use in the Optical Processing Unit (OPU). The OPU performs a non-linear random projection, with *fixed unknown parameters*. A fully-connected layer is then used to obtain a prediction from the output of the OPU. Direct Feedback Alignment is used to train the layer underneath the OPU.

With $\mathbf{U}$ a *fixed* complex Gaussian random matrix of size up to $10^6 \times 10^6$, which entries are not readily known. In the following, we sometimes refer to $\mathbf{U}$ as the *transmission matrix* (TM). The input is binary and the output is in 8 bits.

The matrix $\mathbf{U}$ is physically implemented through the diffusive medium. As only the non-linear intensity $|\mathbf{Ux}|^2$ can be measured, an attacker has to perform *phase retrieval* to retrieve the coefficients of $\mathbf{U}$. We develop such an attack scenario in Section 4.

**Direct Feedback Alignment.** Because the fixed random parameters implemented by the OPU are unknown, it is impossible to backpropagate through it. We bypass this limitation by training layers upstream of the OPU using Direct Feedback Alignment (DFA) (Nøkland, 2016).

In a fully connected network, at layer $i$ out of $N$, neglecting biases, with $\mathbf{W}_i$ its weight matrix, $f_i$ its non-linearity, and $\mathbf{h}_i$ its activations, the forward pass can be written as $\mathbf{a}_i = \mathbf{W}_i\mathbf{h}_{i-1}, \mathbf{h}_i = f_i(\mathbf{a}_i)$. $\mathbf{h}_0 = X$ is the input data, and $\mathbf{h}_N = f(\mathbf{a}_N) = \hat{\mathbf{y}}$ are the predictions. A task-specific cost function $\mathcal{L}(\hat{\mathbf{y}}, \mathbf{y})$ is computed to quantify the quality of the predictions with respect to the targets $\mathbf{y}$. The weight updates are obtained through:

$$\delta\mathbf{W}_i = -\frac{\partial\mathcal{L}}{\partial\mathbf{W}_i} = -[(\mathbf{W}_{i+1}^T\delta\mathbf{a}_{i+1}) \odot f_i'(\mathbf{a}_i)]\mathbf{h}_{i-1}^T \quad (2)$$

where $\odot$ is the Hadamard product. With DFA, the gradient signal $\mathbf{W}_{i+1}^T\delta\mathbf{a}_{i+1}$ of the (i+1)-th layer is replaced with a random projection of the gradient of the loss at the top layer $\delta\mathbf{a}_y$–which is the error $\mathbf{e} = \hat{\mathbf{y}} - \mathbf{y}$ for the cross-entropy loss:

$$\delta\mathbf{W}_i = -[(\mathbf{B}_i\delta\mathbf{a}_y) \odot f_i'(\mathbf{a}_i)]\mathbf{h}_{i-1}^T, \delta\mathbf{a}_y = \frac{\partial\mathcal{L}}{\partial\mathbf{a}_y} \quad (3)$$

**ROPUST** We propose to replace their classifier with the ROPUST module to enhance the adversarial robustness of pretrained robust models (Fig. 2). We use robust models

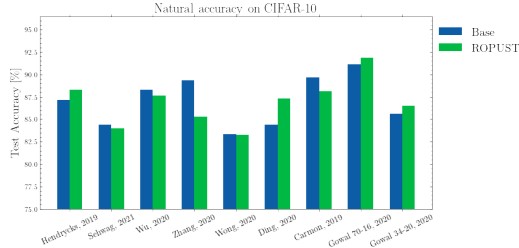

Figure 3. **Our ROPUST defense comes at no cost in natural accuracy.** In some cases, natural accuracy is even improved. The model from Zhang, 2020 (Zhang et al., 2020) is an isolated exception. The papers related to each model are cited in Fig. 1.

from the RobustBench model zoo, extracting and freezing their convolutional stack. The robust convolutional features go through a fully connected layer and a sign function, preparing them for the OPU. The OPU then performs a non-linear random projection, with fixed unknown parameters. The predictions are obtained through a final fully-connected layer. While the convolutional layers are frozen, we train the ROPUST module on natural data using DFA to bypass the non-differentiable photonic hardware.

**Attacking ROPUST.** Previous work has shown that methods devoid of weight transport are not effective in generating compelling adversarial examples (Akrout, 2019). Therefore, we use backward pass differentiable approximation (BPDA) in place of DFA when attacking our defense: we relax non-differentiable layers to a differentiable version. For the binarization function, we simply use the derivative of $\tanh$ in the backward pass, while we approximate the transpose of the obfuscated parameters with a different fixed random matrix drawn at initialization of the module.

## 3. Evaluating ROPUST on RobustBench

All of the attacks are performed on CIFAR-10 (Krizhevsky, 2009), using a differentiable backward pass approximation

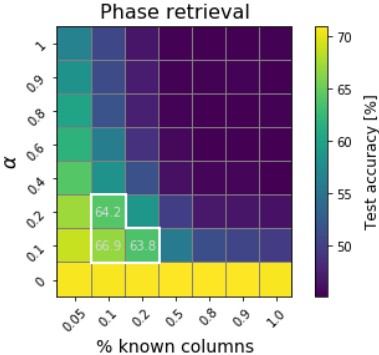

*Figure 4.* **Performance of an APGD-CE attack with a retrieved matrix in place of the, otherwise unknown, transpose of the transmission matrix.** A better knowledge of the transmission matrix correlates with the success of the attack, with a sharp phase transition. It may seem that even a coarse-grained knowledge of the TM can help the attacker. However, even state-of-the-art phase retrieval methods operate only in the white contoured region, where the robustness is still greater than the *Base* models. We highlighted the accuracies achieved under attack in this region.

(Athalye et al., 2018) as explained in Section 2.2. For our experiments, we use OPU input size 512 and output size 8000. We use the Adam optimizer (Kingma & Ba, 2014), with learning rate 0.001, for 10 epochs. The process typically takes 10 minutes on a single NVIDIA V100 GPU.

We show our results on nine different models in Robust-Bench in Fig. 1. The performance of the original pretrained models from the RobustBench leaderboard is reported as *Base*. *ROPUST* represents the same models equipped with our defense. Finally, *Transfer* indicates the performance of attacks created on the original model and transferred to fool the ROPUST defense. For all models considered, ROPUST improves the robustness significantly, even under transfer. For transfer, we also tested crafting the attacks on the *Base* model while using the loss of the ROPUST model for the learning rate schedule of APGD. We also tried to use the predictions of ROPUST, instead of the base model, to *remove* the samples that were successfully attacked from the next stage of the ensemble; however, these modifications did not improve transfer performance. We remark that the robustness increase typically comes at no cost in natural accuracy; we show the accuracy on natural data of the *Base* and the *ROPUST* models in Fig. 3. We ablate our defense against white-box attacks in the supplementary.

## 4. Phase retrieval attack

Our defense leverages parameter obfuscation to achieve robustness. Yet, however demanding, it is still technically possible to recover the parameters through phase retrieval schemes (Gupta et al., 2019; 2020). To provide a thorough and fair evaluation of our attack, we study in this section *phase retrieval* attacks. We first consider an idealized setting, and then confront this setting with a real-world phase retrieval algorithm from (Gupta et al., 2020).

**Ideal retrieval model.** We build an idealized phase retrieval attack, where the attacker knows a certain fraction of columns, up to a certain precision. We model the retrieved matrix $\mathbf{U}'$ as a linear interpolation of the real transmission matrix $\mathbf{U}$ and a completely different random matrix $\mathbf{R}$. In practice, this model is valid only for a certain fraction of columns, and the remaining ones are modeled as independent random vectors. We can model this with a Boolean mask matrix $\mathbf{M}$, so our retrieval model in the end is:

$$\mathbf{U}' = \alpha\mathbf{U} \odot \mathbf{M} + (1-\alpha)\mathbf{R} \qquad (4)$$

In this setting, we vary the knowledge of the attacker from the minimum to the maximum by varying $\alpha$ and the percentage of retrieved columns, and we show how the performance of our defense changes in Fig. 4. In this simplified model only a crude knowledge of the parameters seems sufficient, given the sharp phase transition. We now need to chart where state-of-the-art retrieval methods are on this graph to estimate their ability to break our defense.

**Real-world retrieval performance.** State-of-the-art phase retrieval methods seek to maximize output correlation, i.e. the correlation on $\mathbf{y}$ in Eq. 1, in place of the correlation with respect to the parameters of the transmission matrix, i.e. $\mathbf{U}$ in Eq. 1. We find this is a significant limitation for attackers. In Fig. 4, following numerical experiments, we highlight with a white contour the operating region of a state-of-the-art phase retrieval algorithm (Gupta et al., 2020), showing that it can manage to only partially reduce the robustness of ROPUST.

## 5. Conclusion

We introduced ROPUST, a drop-in module to enhance the adversarial robustness of pretrained already robust models. Our technique relies on parameter obfuscation guaranteed by a photonic co-processor, and a synthetic gradient method: it is simple, fast and widely applicable.

We thoroughly evaluated our defense on nine different models in the standardized RobustBench benchmark, reaching state-of-the-art performance. In light of these results, we encourage to extend RobustBench to include parameter obfuscation methods.

Finally, we developed a new kind of attacks, *phase retrieval attacks*, specifically suited to parameter obfuscation defense such as ours, and we tested their effectiveness. We found that the typical precision regime of even state-of-the-art phase retrieval methods is not enough to completely break ROPUST.

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

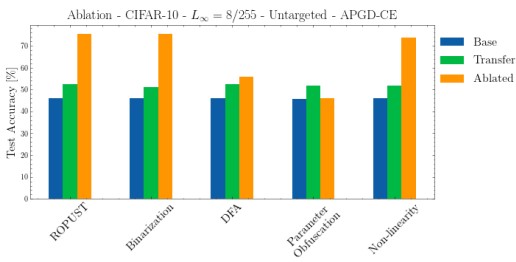

Figure 5. **Removing either parameter obfuscation or DFA from our defense causes a large drop in accuracy.** Robustness is given by the inability to efficiently generate attacks in a white-box settings when the parameters are obfuscated, and DFA is capable of generating partially robust features. Even though the non-linearity $|.|^2$ does not contribute to robustness, it is key to obfuscation, preventing trivial retrieval. Transfer performance does not change much when removing components of the defense. The performance of the `Base` is shown for comparison.

## Appendix

### Description of the attacks in AutoAttack

APGD-CE is a standard PGD where the step size is tuned using the loss trend information, squeezing the best performance out of a limited iterations budget. APGD-T, on top of the step size schedule, substitutes the cross-entropy loss with the Difference of Logits Ratio (DLR) loss, reducing the risk of vanishing gradients. Square attack is based on a random search. Random updates $\delta$ are sampled from an attack-norm dependent distribution at each iteration: if they improve the objective function they are kept, otherwise they are discarded. FAB attack aims at finding adversarial samples with minimal distortion with respect to the attack point. With respect to PGD, it does not need to be restarted and it achieves fast good quality results.

### Ablation study: white-box setting

We perform an ablation study and find that the robustness of our defense against white-box attacks comes from both *parameter obfuscation* and DFA. We use the model from (Wong et al., 2020) available in the RobustBench model zoo. It consists in a PreAct ResNet-18 (He et al., 2016), pretrained with a *"revisited"* FGSM of increased effectiveness. We conduct the ablation study by removing a single component of our defense at a time in simulation: binarization, DFA, parameter obfuscation, and non-linearity $|.|^2$ of the random projection. To remove DFA, we also remove the binarization step and train the ROPUST module with backpropagation, since we have access to the transpose of the transmission matrix in the simulated setting of the ablation study. We show the results in Fig. 5: removing the non-linearity $|.|^2$ and the binarization does not have an effect, with the robustness given by *parameter obfuscation*

and DFA.

### Impact statement

Adversarial attacks have been identified as a significant threat to applications of machine learning in-the-wild. Developing simple and accessible ways to make neural networks more robust is key to mitigating some of the risks and making machine learning applications safer. More robust models would enable a wider range of business applications, especially in safety-critical sectors. We do not foresee negative societal impacts of our work, beyond the risk of our defense being broken by future developments in adversarial attacks. A limit of our work is that we prove increased robustness only empirically and not theoretically. However, theoretically grounded defense methods typically fall short of other techniques more used in practice. We rely on photonic hardware accessible by anyone, similarly to GPUs or TPUs on commercial cloud providers. We performed all of our experiments on single-GPU nodes with NVIDIA V100, and an OPU, on a cloud provider. We estimate a total of $\sim 500$ GPU hours was spent.