# OpenReview forum: "ROPUST: Improving Robustness through Fine-tuning with Photonic Processors and Synthetic Gradients"
_ICML.cc/2021/Workshop/AML — ICML 2021 Workshop AML Poster_

### Official Review · Reviewer_LcfM · 2021-06-19
**The proposed defense method leverages a photonic co-processor for physical parameter obfuscation, which has a certain novelty.**

**Rating:** Accept
**Confidence:** 4

**Review:**

The paper proposed a new defense method called "ROPUST", which leverages a photonic co-processor for physical parameter obfuscation.  Besides, it proposed a phase retrieval attack to test the robustness of the proposed defense. The experimental results show that, with the proposed defense mechanism， the classification accuracy of the tested nine SOTA defense models against AutoAttack has been improved significantly.

pros:
1. The writing is good and easy to follow.
2. The paper introduces the Optical Processing Unit (OPU) for physical parameter obfuscation, which has a certain novelty.
3. The experimental results show the effectiveness of the proposed defense method and robustness to the new proposed  phase retrieval attack.

cons:
1. Maybe it's better to set up some comparative experiments with other gradient obfuscation based defense methods.

---

### Decision · Program_Chairs · 2021-06-21

**Decision:**

Accept (Poster)

**Comment:**

This paper proposed a new defense method "ROPUST". The robustness has been improved significantly. More experiments can be conducted to further test against gradient obfuscation.